# Mathematical Model of Fuse Effect Initiation in Fiber Core

Victoria A. Starikova [1], Yuri A. Konin [1,2,*], Alexandra Yu. Petukhova [1], Svetlana S. Aleshkina [3,*], Andrey A. Petrov [2] and Anatolii V. Perminov [1]

[1] Applied Mathematics and Mechanics Faculty, Perm National Research Polytechnic University, 614990 Perm, Russia; scherbackova.vict@mail.ru (V.A.S.); umalia.ookami.98@mail.ru (A.Y.P.); perminov1973@mail.ru (A.V.P.)
[2] Institute of Laser Technologies, ITMO University, 197101 St. Petersburg, Russia; aapetrov@itmo.ru
[3] Dianov Fiber Optics Research Center, Prokhorov General Physics Institute of the Russian Academy of Sciences, 119333 Moscow, Russia
[*] Correspondence: yuri-konin@yandex.ru (Y.A.K.); sv_alesh@fo.gpi.ru (S.S.A.)

**Abstract:** This work focuses on the methods of creating in-fiber devices, such as sensors, filters, and scatterers, using the fiber fuse effect. The effect allows for the creation of structures in a fiber core. However, it is necessary to know exactly how this process works, when the plasma spark occurs, what size it reaches, and how it depends on external parameters such as power and wavelength of radiation. Thus, this present study aims to create the possibility of predicting the consequences of optical breakdown. This paper describes a mathematical model of the optical breakdown initiation in a fiber core based on the thermal conductivity equation. The breakdown generates a plasma spark, which subsequently moves along the fiber. The problem is solved in the axisymmetric formulation. The computational domain consists of four elements with different thermophysical properties at the boundaries of which conjugation conditions are fulfilled. The term describing the heat source in the model is determined by the wavelength of radiation and the refractive indices of the core and the shell and also includes the radiation absorption on the released electrons during the thermal ionization of the quartz glass. The temperature field distributions in the optical fiber are obtained. Based on the calculations, it is possible to estimate the occurrence times of various phase states inside the fiber, in particular, the plasma spark occurrence time.

**Keywords:** optical fiber; fuse effect; optical breakdown; thermal conductivity equation; mathematical modeling

## 1. Introduction

The fiber fuse effect, or an optical discharge, is a phenomenon that occurs in fiber-optic lines during the propagation of high-power density radiation. The fiber fuse effect is a multistage process that begins with the optical breakdown of the fiber end face and then continues as a plasma spark moving along the fiber towards the radiation source, which destroys large sections of fiber-optic lines (Figure 1). This effect was first described by Kashyap and Blow in 1987–1988 [1–3].

At present, a study of the optical breakdown initiation and plasma spark motion along fibers is one of the urgent tasks in the development field of fiber-optic instruments. Of particular interest in this connection are quasiperiodic structures resulting from the passage of plasma spark. These structures can be used as sensing elements for sensors or radiation scatterers.

There are several papers related to the phenomenon of the fuse effect, which describe its physical essence and consequences for optical fibers, as well as methods for its study. In [4–6], optical discharge in light guides and its interaction with the fiber material are described. The mechanism of the optical breakdown initiation in a fiber light guide is closely related to the nonlinear absorption of quartz glass [6–8]. If the power density

is sufficient to sustain the optical breakdown, the resulting plasma focus flares up and begins to propagate along the fiber core [9,10]. The works of Y. Shuto [7–9,11] studied the phenomenon of the fuse effect, leading to the destruction of optical fibers under the influence of high-power laser pulses. The article [10] describes the formation of cavities in the fibers and their dynamics in the propagation process of plasma spark along the fiber.

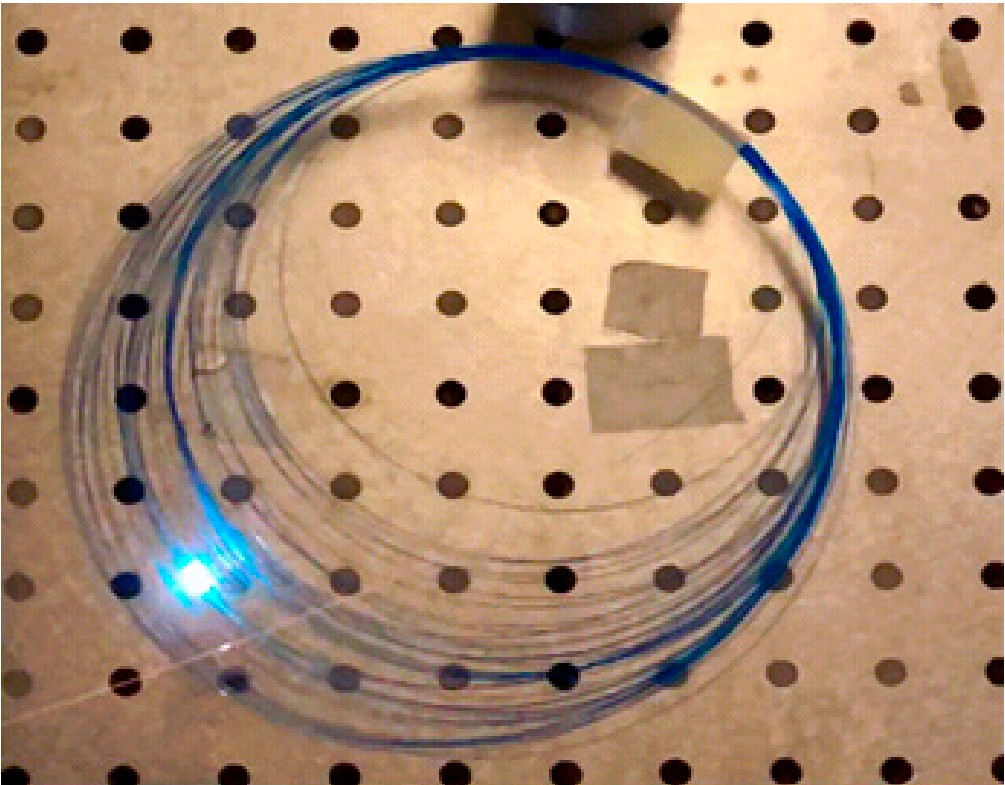

**Figure 1.** Plasma spark moving in an optical fiber.

In general, the most significant contribution to the study of the fiber fuse effect at the fundamental level was made by the works of Y. Shuto [7–9,11] as well as S. Todoroki [10–13]. The latter devoted his research to the experimental confirmation of theoretical developments. The most interesting article of S. Todoroki in the context of this study is the work [10], in which the initiation times of the fuse effect were experimentally investigated. The minimum reaction initiation time was 0.18 s, and the minimum power at which it was possible to start the reaction in the SMF-28 fiber was 0.4 W [10].

Articles [11,12] are devoted to the study of optical breakdown initiation in the presence of an electric discharge or contamination of optical fiber connectors. The cited sources deepen our understanding of the fiber fuse phenomenon and may be useful for developing measures to prevent or manage it.

The dimensions of the internal quasiperiodic structure of an optical fiber obtained during the optical discharge passage depend on the radiation power, its mode composition, and the fiber geometry [14,15]. As mentioned above, these structures can be used as a sensing element for a fiber optic sensor based on a Fabry–Perot interferometer or a fiber optic radiation scatterer [16,17]. For different sensors and optical radiation scatterers, there are different requirements for the geometrical parameters of the internal structure [18,19].

Our team is developing technology for creating in-fiber devices such as sensors, filters, and radiation scatterers that exceed current fiber device technologies in cost effectiveness, speed of device creation, and improved durability of the resulting devices. However, different devices require different internal structures formed during fiber fuse. Therefore, it is very important to know how this process works, when the plasma spark arises, what sizes it reaches, and how it depends on external parameters such as the power and wavelength

of radiation. Thus, this present study is aimed at creating a possibility to predict the consequences of optical breakdown.

This present work is devoted to the study of the thermal physical characteristics of optical fibers using mathematical modeling of the optical breakdown initiation process in the fiber. This stage is shown in Figure 2. Thanks to the mathematical modeling of the fuse effect, it is possible to better understand the causes and mechanisms of its occurrence and to develop methods for controlling this phenomenon.

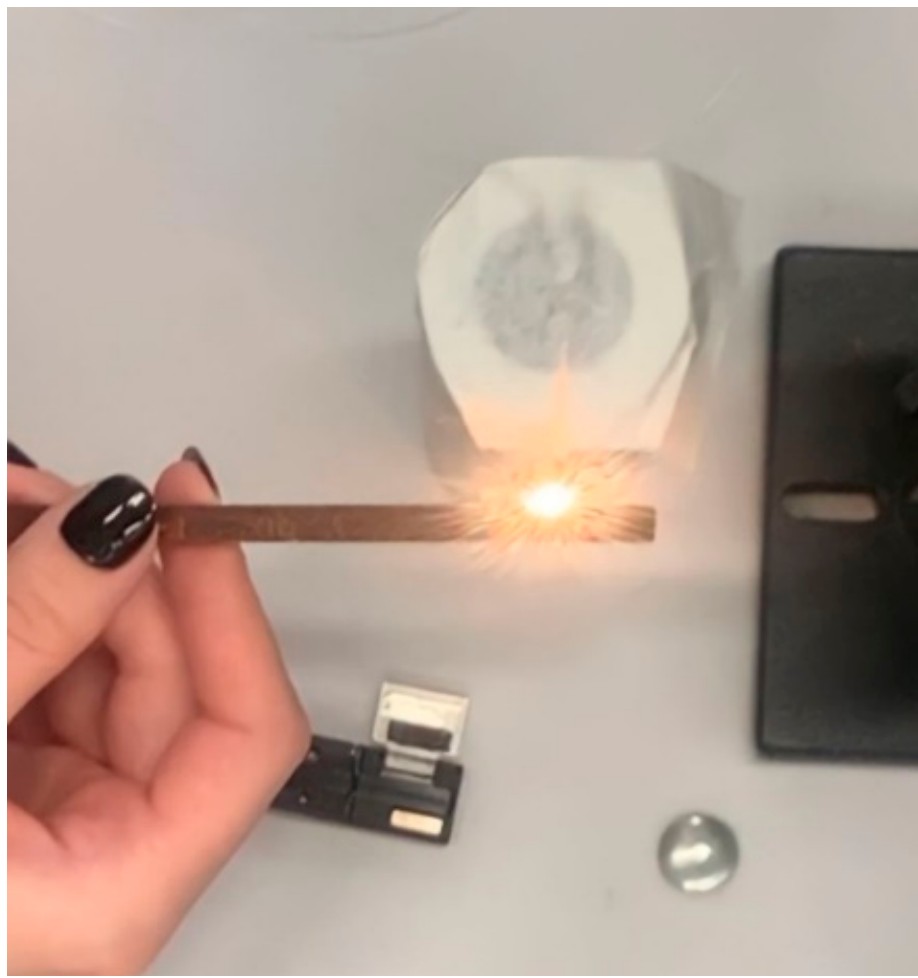

**Figure 2.** The moment of plasma spark initiation in the fiber.

This article presents a relatively simple mathematical model of the initial fiber fuse stage based on the thermal conductivity equation. The model investigates the temperature field distribution in the fiber, which is considered a solid, i.e., phase transitions are neglected. As a result of the breakdown, a plasma spark is produced. The process of plasma spark motion along the fiber was not considered at this stage. All calculations were performed in the COMSOL Multiphysics software package.

## 2. Mathematical Model of Plasma Spark Initiation

We study an anisotropic optical fiber, which has a cylindrical shape, so cylindrical coordinates are used for modeling, where the z-axis is directed along the fiber axis and the r-axis along its radius. The computational domain is presented in Figure 3. The problem was solved in the axisymmetric formulation, so all calculations were performed in the z–r plane in half of the fiber and metal plate. At the fiber symmetry axis (r = 0), the solution symmetry conditions were set. The temperature field independent of the polar angle was determined within the model. The computational domain in Figure 3 consists of five

elements with different thermophysical properties. At the interfaces of the elements, the thermal balance conditions are satisfied. They will be discussed below.

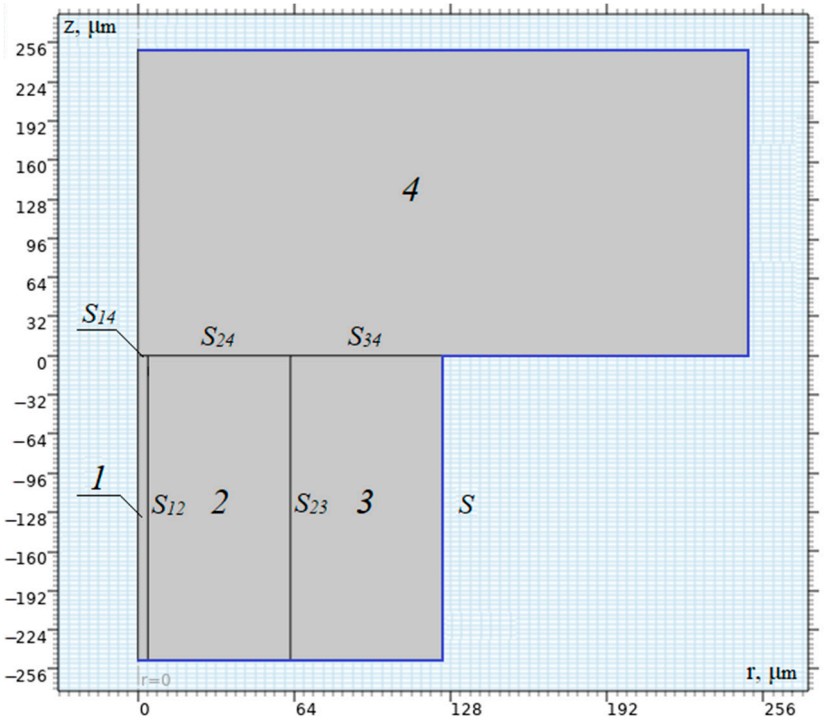

**Figure 3.** Schematic diagram of the computational domain in axisymmetric formulation, consisting of a core (*1*), a shell (*2*), a protective coating of optical fiber (*3*), and a metal plate—the initiator of optical breakdown (*4*).

A section of a single-mode fiber with a stepped refractive index profile SMF-28e (see Figure 3), consisting of a core (*1*), a shell (*2*), and a protective coating (*3*), is studied. The fiber is tightly pressed against the initiator, in this case, a duralumin plate (*4*). The initial and ambient temperatures of the fiber and the metal plate are assumed to be $T_0$ = 293 K.

In the calculations, the physical properties of the optical fiber were set according to the specifications of the fiber SMF-28e. Duralumin plate D16 was used according to the 2024 International Organization for Standardization (ISO). The materials and dimensions of domains of the calculation area are presented in Table 1. Thermal and optical parameters of materials were set for each domain of the computational area and selected from the COMSOL Multiphysics library.

**Table 1.** Materials and dimensions of elements of the calculation area.

| Domain | Material | z, μm | r, μm |
|---|---|---|---|
| A fiber core | Ge- doped $SiO_2$ (fused quartz) [solid, average] | 250 | 4.1 |
| A fiber shell | $SiO_2$ (fused quartz) [solid, average] | 250 | 58.4 |
| A protective coating | PMMA—Polymethyl methacrylate | 250 | 62.5 |
| A duralumin plate | duralumin 2024 [solid] | 250 | 250 |

The fiber core (*1*), its shell (*2*), the protective strengthening coating (*3*), and the duralumin plate (*4*) in Figure 3 have contact boundaries between them at $S_{12}$, $S_{14}$, $S_{23}$, $S_{24}$, and $S_{34}$. The outer boundary of the fiber and the metal plate with the environment is denoted by $S$.

Within the framework of the mathematical model, the distribution of the temperature field T $(r, z, t)$ in the entire computational domain (Figure 3) is described via the following heat conduction equation [20]:

$$\rho_i C_{Pi} \frac{\partial T}{\partial t} = k_i \left( \frac{\partial^2 T}{\partial r^2} + \frac{1}{r} \frac{\partial T}{\partial r} + \frac{\partial^2 T}{\partial z^2} \right) + Q, \tag{1}$$

where $i$ = 1, 2, 3, 4 is the number of the domain in the computational area (Figure 3), $\rho_i$ is densities [kg/m$^3$], $C_{Pi}$ is specific heat capacities [J/(kg·K)], $k_i$ is thermal conductivity of materials [W/(m·K)], and $Q$ is heat source, the intensity of which is determined via the optical radiation [W/m$^3$]. These functions have a complex form depending on the temperature and are shown in Appendix A, Table A1. As mentioned above, all thermophysical properties of materials are taken from the COMSOL Multiphysics library.

Assuming that the temperature at the transition across the interface changes continuously, at the borders $S_{12}$, $S_{14}$, $S_{23}$, $S_{24}$, and $S_{34}$ of the contiguous domains of the calculation scheme, the contiguity or balance of heat fluxes conditions apply:

$$\left( k_i \frac{\partial T}{\partial r} l_r + k_i \frac{\partial T}{\partial z} l_z \right)_{S_{ij}} = \left( k_j \frac{\partial T}{\partial r} l_r + k_j \frac{\partial T}{\partial z} l_z \right)_{S_{ij}} \tag{2}$$

where $l_r$ and $l_z$ are the projections of the unit vector of the normal to the corresponding surfaces on the coordinate axes $z$ and $r$. The outer boundary of the computational domain $S$ was assumed to be perfectly conductive, and the ambient temperature was set on it at $T_0$ = 293 K.

The last term of the thermal conductivity equation $Q$ describes the heat loss that occurs due to the interaction of optical radiation propagating along the fiber with the contact boundary between the end of the fiber and the metal plate. The contact between the fiber face and the plate is assumed to be ideal. After reflection, a standing wave is generated in the core, the energy of which is absorbed by the substance near the contact boundary. The intensity of this heat source depends on the optical power and the wavelength of radiation entering the optical fiber and is calculated using the following formula:

$$Q(r,z) = \alpha(r, \; z) \frac{P}{A_{eff}} \cdot \Gamma(r,z) \tag{3}$$

where $\alpha = (\alpha_0 + \alpha_e)$ is fiber absorption coefficient [m$^{-1}$], $P$ is input power [W], $\Gamma$ is standard Gaussian distribution $\Gamma(x) = \frac{1}{\sqrt{2\pi}} \cdot e^{-\frac{x^2}{2}}$, and $A_{eff}$ is effective area of the mode spot [m$^{-2}$]. Because the problem is solved as a thin 2D layer, the standard Gaussian distribution takes a 2D form as follows:

$$\Gamma(r,z) = \Gamma(r) \cdot \Gamma(z) \tag{4}$$

The absorption coefficient $\alpha$ is the sum of the absorption coefficient under normal conditions (at temperatures close to 293 K) $\alpha_0$ and the absorption coefficient of electron gas $\alpha_e$ appearing at temperatures above 1273 K. The coefficient $\alpha_0 \sim 1$ m$^{-1}$ [7] describes nonlinear absorption of radiation on chaotically located in the fiber defects like SiE′ or GeE′. Such defects or XE′-centers (Ge germanium or Si silicon can be instead of X) are connected with formation of oxygen holes in bonds: Si-X → Si-O-X. Defects appear in the core of the fiber as a result of chemical reactions during its manufacture [20,21].

When heated, the concentration of GeE′ centers remains constant, while at temperatures exceeding 1000 K, the concentration of SiE′ decreases to zero [11]. When the core temperature reaches 1273 K, thermal ionization of GeE′ centers begins [21]. As a result, electron gas, i.e., a large number of free electrons, is formed in the core of the fiber. The concentration of the released electrons coincides with the concentration of GeE′-centers ions.

According to [22], in a fiber doped with 4 mol% germanium, the concentration of GeE′-centers ions is calculated as follows:

$$n_{\text{GeE}'} = n_p \exp\left(-\frac{E_f}{k_B T}\right),$$ (5)

where $E_f$ = 2.5 eV is formation energy GeE′, $n_p$ = 1.72 × $10^{15}$ m$^{-3}$ is concentration GeE′centers at normal temperature, and $k_B$ is Boltzmann constant. The conductivity of quartz due to free electrons will be expressed as follows:

$$\sigma = e\mu_e n_e = e\mu_e n_p \exp\left(-\frac{E_f}{k_B T}\right),$$ (6)

where $e$ is electron charge modulus, and $\mu_e$ is the drift mobility of electrons. The latter varies from 0.7 to 6.3 × $10^{-3}$ m$^2$/(V·s) in the calculations; this value was assumed to be 5 × $10^{-3}$ m$^2$/(V·s) [8], with electron concentration $n_e = n_{\text{GeE}'}$. Taking into account (5), the absorption coefficient of electron gas in the fiber from Formula (3) will be equal to the following:

$$\alpha_e = \frac{k_B \cdot n_1}{\sqrt{2}}\left[\sqrt{1 + \left(\frac{\mu_0 c_0 \sigma}{k_0 n_1^2}\right)^2} - 1\right]^{\frac{1}{2}} \sim \frac{\mu_0 c_0 \sigma}{2n_1} = \frac{\mu_0 c_0}{2n_1} e\mu_e n_p \exp\left(-\frac{E_f}{k_B T}\right),$$ (7)

where $n_1$ is refractive index in the fiber core, $\mu_0$ is magnetic constant, $c_0$ is speed of light in a vacuum, and $k_0$ is wave number in vacuum.

The effective area of the mode spot in (3) is calculated using the effective radius for each of the wavelengths as follows [23]:

$$A_{eff} = \pi a^2 \left(0.65 + \frac{1.619}{V^{\frac{3}{2}}} + \frac{2.879}{V^6}\right)^2,$$ (8)

where $a$ is the radius of the fiber core (see Table 1). Parameter $V$ is normalized frequency, determines the number of transported modes of a given radiation in a given optical fiber geometry, and is calculated using the following formula:

$$V = \frac{2\pi a}{\lambda}\sqrt{n_1^2 - n_2^2},$$ (9)

where $\lambda$ is wavelength of transported radiation; $n_1$ is refractive index of the fiber core; $n_2$ is refractive index of the fiber shell.

Table 2 shows the values of wavelengths and refractive indices of the core and fiber cladding, for which the calculations were performed. Table 2 also shows the values of the parameters $V$ and $A_{eff}$, calculated according to Formulas (8) and (9).

**Table 2.** Values of wavelengths and refractive indices of the core and fiber cladding.

| $\lambda$, nm | $n_1$ | $n_2$ | $V$ | $A_{eff}$, µm$^2$ |
|---|---|---|---|---|
| 1080 | 1.4483 | 1.4439 | 2.76 | 13.25 |
| 1310 | 1.4552 | 1.4508 | 2.17 | 15.29 |
| 1550 | 1.4617 | 1.4573 | 1.88 | 17.27 |
| 2050 | 1.4662 | 1.4618 | 1.39 | 23.49 |

## 3. The Solution Algorithm

The optical fiber industry and fiber-optic devices require science-based technologies. These technologies help to produce, model, and predict future devices and processes. The development of computational methods makes it possible to improve the fiber drawing

process, optimize process parameters, and determine stable drawing modes [24]. A better understanding of the optical breakdown process in the fiber will make it possible to create intrafiber devices with reasonably good repeatability of the characteristics. For example, the article [25] describes methods for modeling the spectrum structure of a Bragg reflector in a fiber. Alternatively, it will improve the signal processing methods in them [26].

While creating a mathematical model, the choice of a simulation tool is also important. In [27], the authors performed a comparison of two application software packages, Matlab and COMSOL Multiphysics, for solving parabolic integro-differential equations, which are used in this present work. The studies have shown that in spite of the limitations of the built-in calculation results' processing functions, the COMSOL Multiphysics software package has an important possibility to modernize the built-in algorithms and calculation methods, for example, by introducing analytical functions, scripts, etc.

The algorithm for solving the presented problem was implemented in the application package Comsol Multyphisycs, which uses the finite element method to solve various physical and engineering problems. To solve our problem, we used the module The Heat Transfer in Solids (ht) Interface. All calculations are performed in the flat axisymmetric formulation.

The computational area consisted of four domains with different thermophysical properties (see Figure 3). In the entire computational domain, except for the region near the $S_{14}$ boundary, where intense heating occurs, a computational grid with elements of the same size was introduced. A small region with linear dimensions equal to the radius of the fiber core was chosen near the $S_{14}$ boundary, where the computational mesh was refined to achieve the best resolution of the temperature field. The general view of the computational grid is shown in Figure 4.

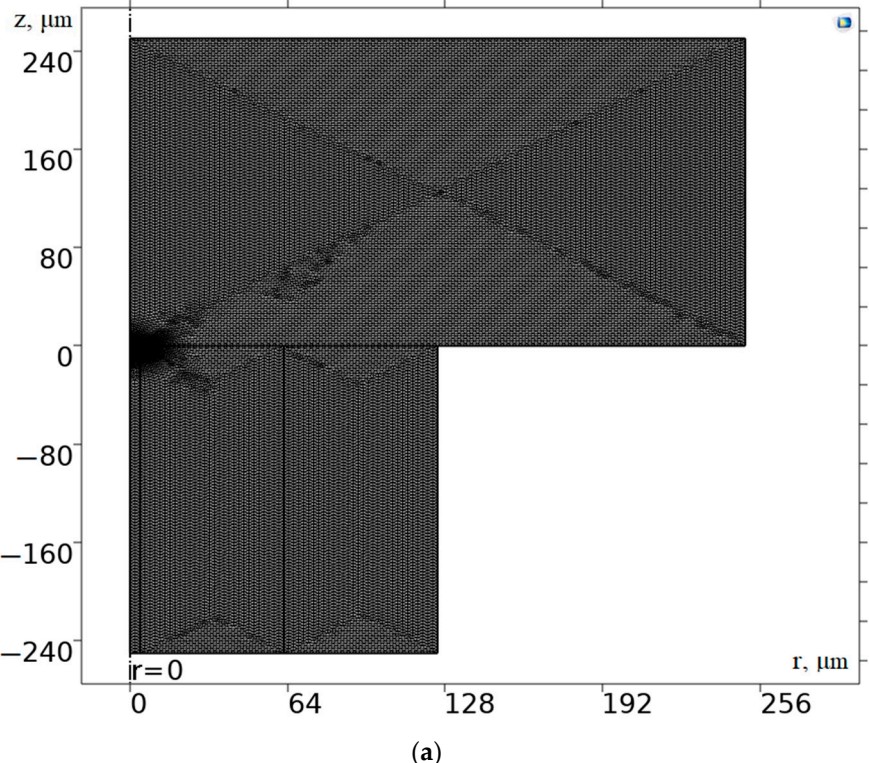

(a)

**Figure 4.** *Cont.*

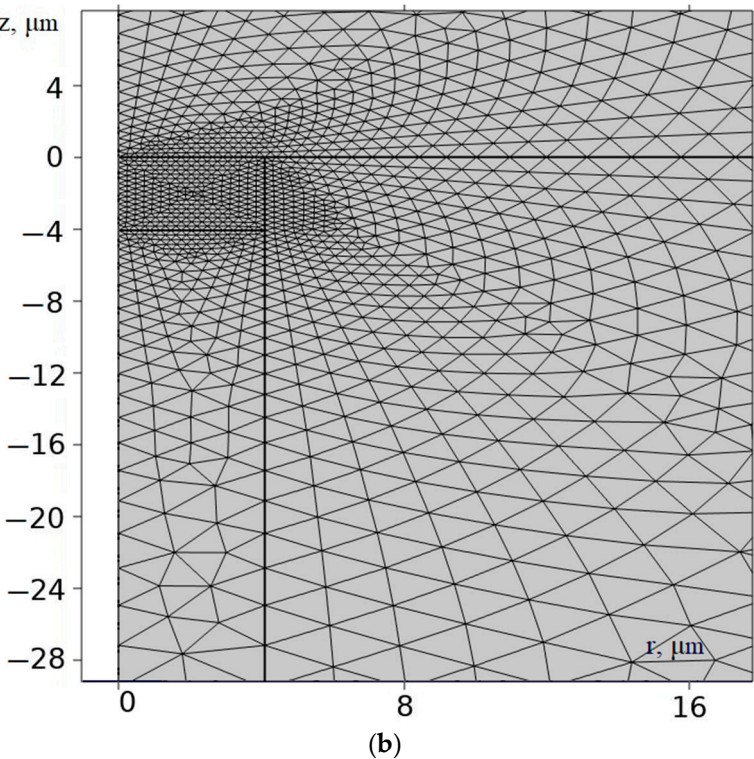

**(b)**

**Figure 4.** View of the computational grid: (**a**) general view; (**b**) view near the heat source.

Fourier's law is satisfied for the heat flux at the boundaries of the elements of the computational grid as follows:

$$q = -k\nabla T, \tag{10}$$

where $k$ is the thermal conductivity of the material, $q$ is the heat flux vector, and $\nabla T$ is the temperature gradient.

The temperature front at each time iteration is interpolated using a linear Lagrange polynomial. Equation (3) is used to calculate the heat source that occurs in the core of the optical fiber. The mathematical model takes into account the distribution of the radiation power along the longitudinal and transverse coordinates. The result is a two-dimensional array of heat source values in the computational domain.

In the entire computational region (Figures 3 and 4), except for the region near the $S_{14}$ boundary, where intensive heating takes place, a computational mesh with the same size elements of 2.5 μm was introduced. A small region with linear dimensions equal to the fiber core radius was isolated in the vicinity of the $S_{14}$ boundary, where the computational mesh was milled for the best resolution of the temperature field.

In the course of test calculations, the convergence of the results was investigated when the number of mesh elements was increased. For this purpose, the dependence of the maximum temperature $T_{max}$, which is reached in the fiber after 1 ms, on the number of mesh elements in the computational domain was plotted. This dependence is shown in Figure 5. The black squares show the temperature values at different numbers of mesh elements, and the red line is the trend line constructed on the basis of the least-squares method. Figure 5 shows that the solution stabilizes around the $T_{max} \approx 177{,}200$ K with a total number of grid elements of more than 15,000. At the same time, the minimum grid cell size was 0.03 μm, which allows us to perform calculations within the continuum approximation.

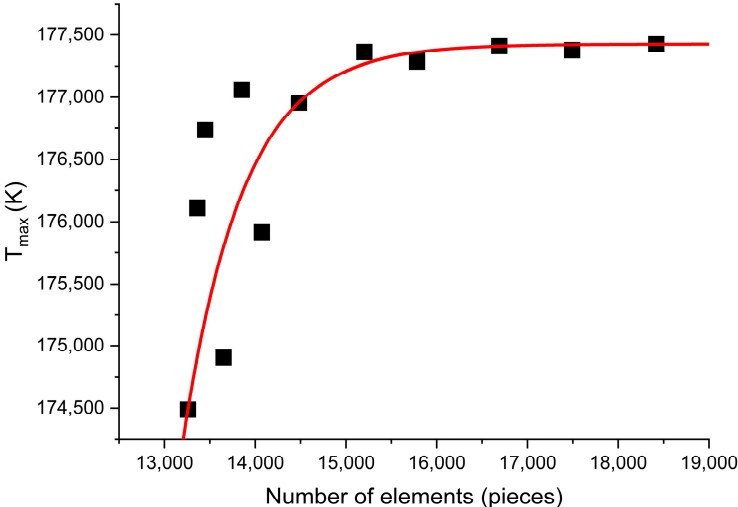

**Figure 5.** Dependence of the maximum temperature in the calculated region in 1 ms on the total number of mesh elements. Black squares indicate temperature values at different number of mesh elements, and red line is trend line.

Figure 6 shows a block diagram of the numerical algorithm, which shows the steps to solve the problem.

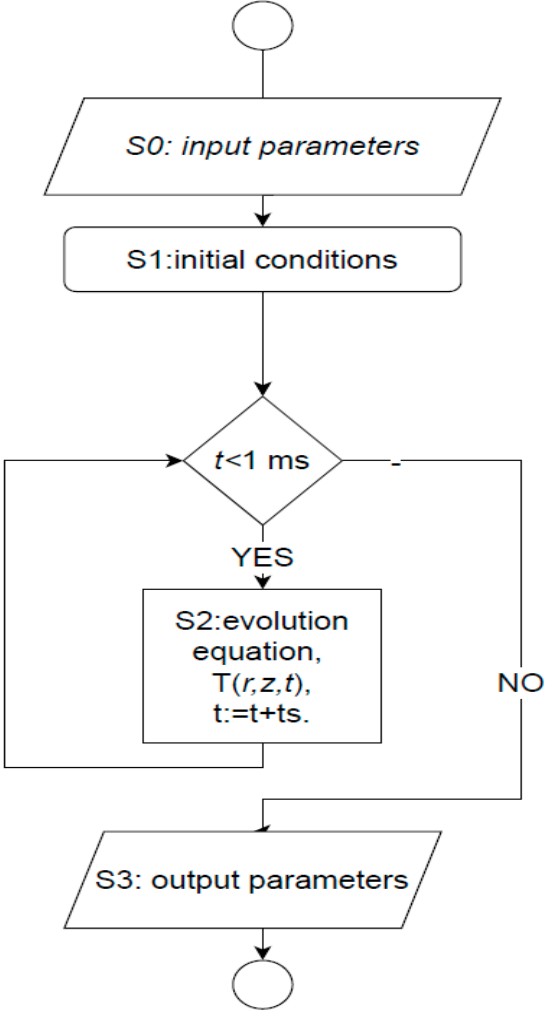

**Figure 6.** Algorithm scheme representation.

Zero step. Before starting the main calculation procedure, a number of physical parameters and constants were set, which were not changed in the course of a particular numerical calculation. A complete list of these parameters and constants is presented in Table 3.

**Table 3.** Values of constant parameters of mathematical model.

| Description | Value |
| --- | --- |
| Boltzmann's constant | $1.38 \times 10^{-23}$ J/K |
| Core refractive index | 1.4617 |
| Charge of electron | $1.6 \times 10^{-19}$ C |
| Concentration | $1.72 \times 10^{15}$ m$^{-3}$ |
| Input power | $1.10 \times$ W |
| Formation energy GeE$'$ | $4.0054 \times 10^{-19}$ J |
| Light velocity | $3 \times 10^8$ m/s |
| Core radius | $4.1 \times 10^{-6}$ m |
| Core layer thickness | $4.1 \times 10^{-6}$ m |
| Magnetic constant | $1.2566 \times 10^{-6}$ H/m |
| Drift mobility of electrons | $5 \times 10^{-3}$ m$^2$/(V·s) |

The first step is the initial conditions. As an initial approximation, a homogeneous temperature field with a value of 293 K was set. Using Formula (3), the initial power distribution of the heat source is calculated. For this purpose, the effective area of the mode spot $A_{eff}$ was calculated using Formulas (8) and (9), and the initial distribution of the absorption coefficient $\alpha_e$ was calculated using Formula (7). The initial time $t$ condition is set to zero, and the time step $ts$ is 0.01 ms.

The second step was to solve the evolution Equation (1). At each time step, the heat fluxes between the domains of the computational domain were calculated using Formula (2), and the heat source capacity was recalculated using Formulas (7)–(9) and (3). A constant temperature of 293 K was set for the outer boundaries of the computational domain. The numerical solution of Equation (1) was performed with a variable time step, which was selected on the basis of the stability conditions of the computational algorithm.

The third step is the output and processing of calculation results.

The algorithm for solving the problem was implemented in the application software package Comsol Multyphisics. The approximations of the thermal conductivity equation, interfacial boundary conditions, and boundary conditions were performed using the finite element method. The authors used standard computational modules, which are implemented in Comsol. The computation time was 22 s with the following system parameters: Intel Core i9-9900K Processor, 64 GB RAM, 1 Tb SSD. Calculations involved 8 cores, 1.5 GB of physical memory, and 1.6 GB of virtual memory. PARDISO Solver was used to solve the system of linear equations in multi-threaded mode. We used the constant Newton's method of solving linear equations. For searching and solving nonlinear equations at each time step Jacobian update was used with a maximum number of iterations equal to 40. To accelerate the convergence of nonlinear equations Anderson acceleration, the following parameters were used: dimension of iteration space = 5, mixing parameter = 0.9, iteration delay = 1.

## 4. Results

During mathematical simulation on the basis of Equations (1)–(10), the temperature field in the optical fiber and duralumin plate was calculated, and the boundaries of temperature fronts corresponding to the regions where various phase states of quartz glass can occur, namely liquid, gaseous and plasma, were determined. The melting temperature of quartz (optical fiber) corresponds to 1440 K, vaporization in the fiber corresponds to 2706 K, and plasma formation starts at 5000 K.

Figure 7 shows an example of the calculated temperature field for radiation with wavelength $\lambda_0 = 1550$ nm and power $P = 1$ W in a time interval of 1 s after the introduction

of radiation. The time interval was chosen from the experimental data [13]. As seen in Figure 7, the temperature front of the plasma formation comes out in the region of the fiber shell but does not reach the protective-strengthening coating.

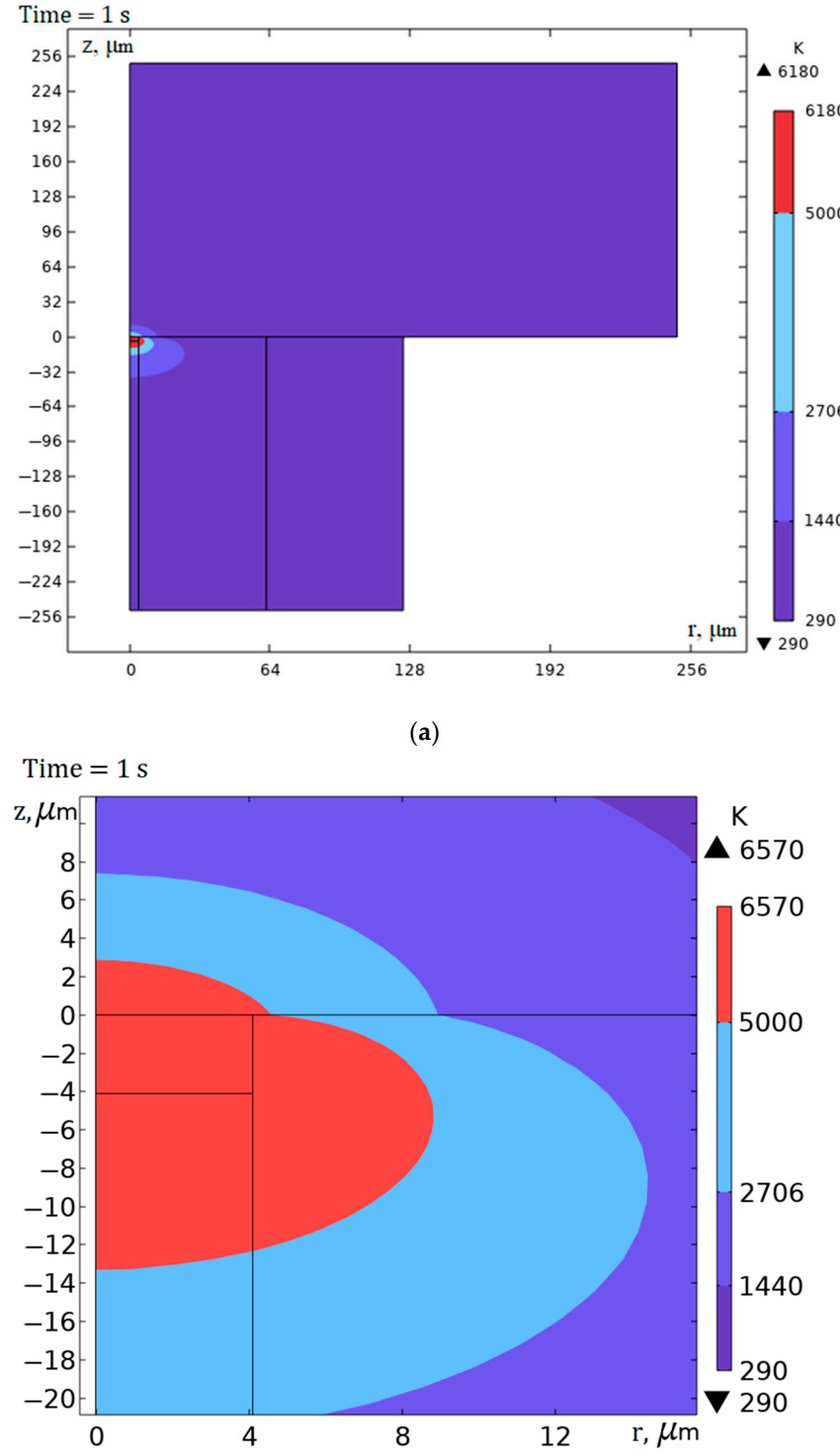

**Figure 7.** Temperature field distribution in the axisymmetric setting (symmetry axis r = 0) in 1 s after introducing radiation into the fiber: (**a**) general view; (**b**) near the heating area.

Experiments show that the size of the plasma spark moving along the fiber and the defects resulting from this motion do not exceed the size of the fiber core [18]. This circumstance suggests that optical breakdown, as a process of the plasma spark motion,

begins at the moment when the temperature front of the plasma formation reaches the core–shell $S_{12}$ boundary. Within the framework of the constructed model, it is possible to estimate the time of occurrence of the optical breakdown $t_{ob}$ for different wavelengths and powers. Plots of the optical breakdown time dependences on the input optical power for four wavelengths are shown in Figure 8.

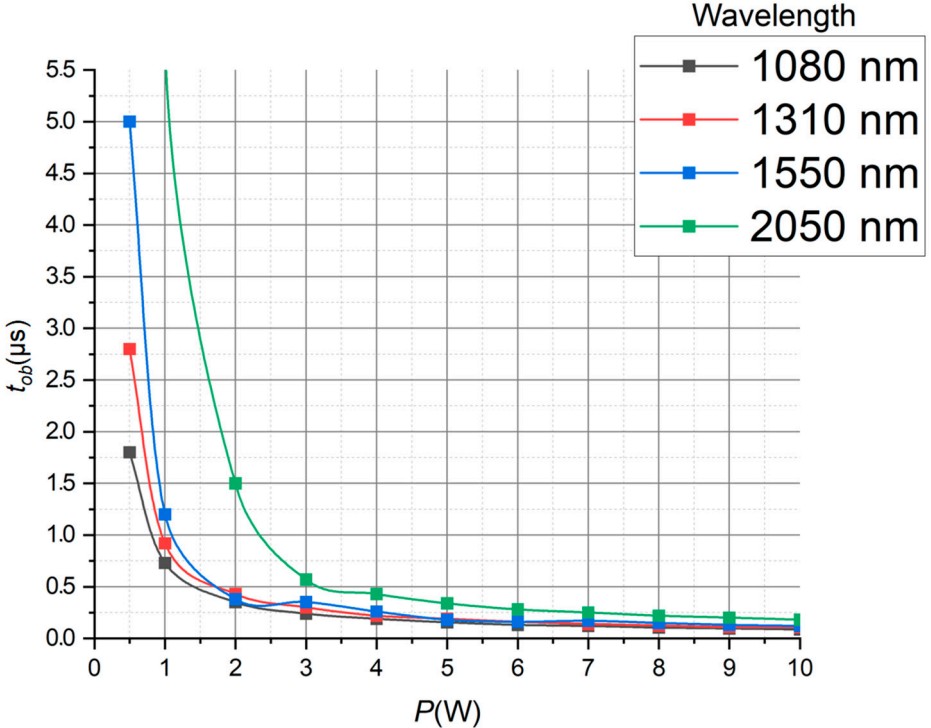

**Figure 8.** Dependences of the optical breakdown time on the input optical power for four wavelengths.

As seen in Figure 8, the time required for the initiation of optical breakdown in the fiber decreases with increasing power according to the law close to the exponential one. At high radiation powers ($P_{max}$ = 10 W), the plasma spark initiation time lies in a narrow time range from 0.09 μs for $\lambda_0$ = 1080 nm to 0.18 μs for $\lambda_0$ = 2050 nm. At low radiation powers, the initiation time increases with increasing wavelength.

As a result of the calculations for different wavelengths, the minimum $P_{min}$ power values necessary to initiate optical breakdown were estimated. The results of the calculations are shown in Table 4, which shows that the minimum power $P_{min}$ required to initiate optical breakdown increases with increasing wavelength $\lambda$. At low powers, there is a significant difference in the initiation times (see Figure 8).

**Table 4.** Minimum power required to initiate optical breakdown.

| $\lambda$, nm | 1080 | 1310 | 1550 | 2050 |
|---|---|---|---|---|
| $P_{min}$, mW | 270 | 335 | 400 | 630 |
| $T_{max}$, K | 14,685 | 14,671 | 14,630 | 14,477 |

At minimum powers, the plasma boundary reaches the border of the fiber core in 1 ms and does not move further. Figure 9 shows the temperature field in the fiber for radiation with wavelength $\lambda_0$ = 1550 nm and power $P_{min}$ = 400 mW. The temperature front of plasma formation does not cross the boundary $S_{12}$ core–shell of the fiber.

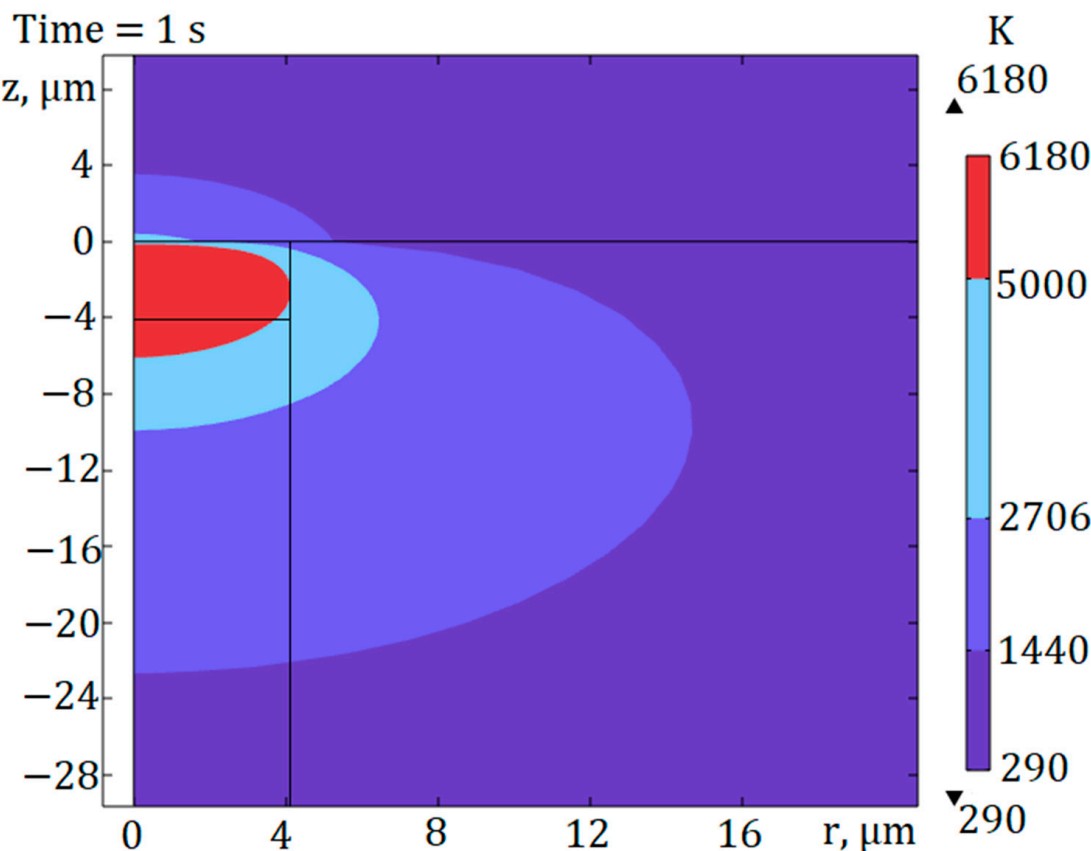

**Figure 9.** Temperature field distribution for radiation with wavelength $\lambda_0$ = 1550 nm and power $P_{min}$ = 400 mW in 1 s from the moment of radiation water.

Figure 10 shows a graph of the maximum temperature $T_{max}$ at the center of the plasma spark initiated in the fiber core as a function of the injected optical power after 1 s from the start of the input for four wavelengths. It can be seen that $T_{max}$ increases linearly with increasing optical radiation power and decreases with increasing wavelength. Table 4 shows the values of maximum temperatures for the minimum powers of radiation initiating the fiber fuse effect. It can be seen that for all wavelengths and powers, this temperature is close to 14,500 K. In Figure 10b, this threshold temperature is indicated by the dotted line.

The results of numerical calculations presented in this paper and in the works of other authors [7–9] have shown that significant temperatures of the order of tens or even hundreds of thousands of kelvins are reached during optical breakdown inside the fiber core. However, it should be noted that such high-temperature values are localized in the center of the plasma region. Figure 11 shows an example of the temperature field for radiation with a wavelength of $\lambda_0$ = 1550 nm and power $P$ = 2 W after 1 ms of the introduction of radiation.

The maximum temperature in the center of the plasma spark after 1 ms for radiation, with wavelength $\lambda_0$ = 1550 nm and power $P$ = 2 W, was 71,000 K. These values qualitatively agree with the results of calculations [8], where the maximum temperature inside the fiber reached 450,000 K for radiation with wavelength $\lambda_0$ = 1550 nm and power $P$ = 2 W in 6 ms after the introduction of radiation.

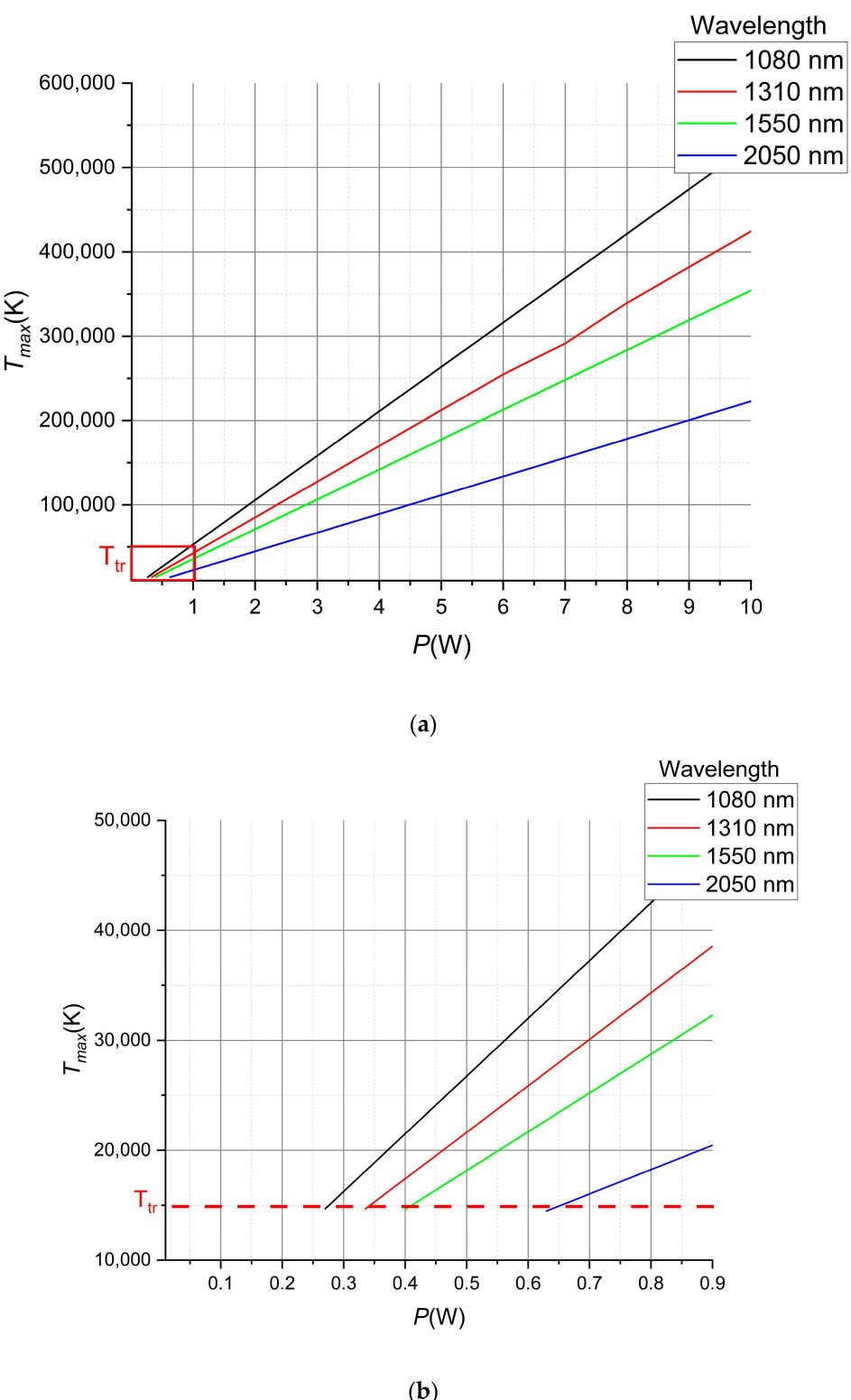

**Figure 10.** Dependence of maximum temperature on radiation power: (**a**) general graph and (**b**) fragment in the area of low radiation power.

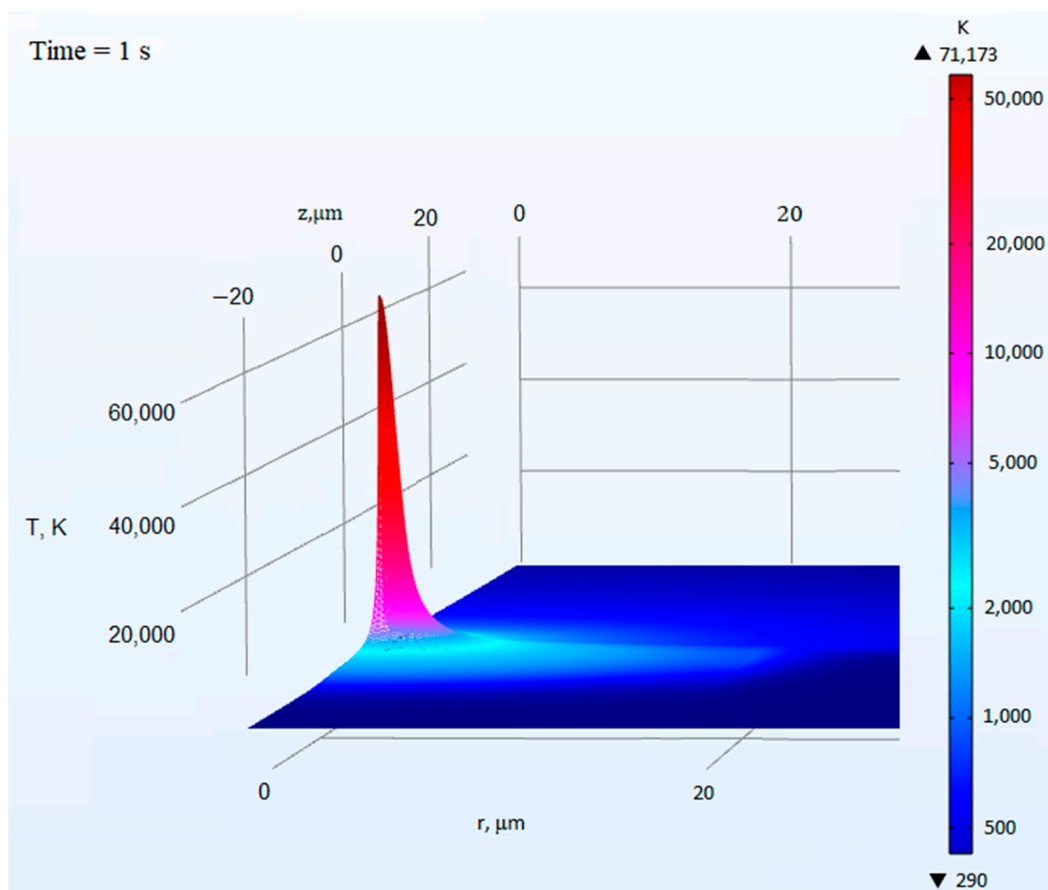

**Figure 11.** Temperature distribution in the center of the plasma spark for radiation with wavelength $\lambda_0$ = 1550 nm and power $P$ = 2 W in 1 ms from the moment of radiation water.

## 5. Discussion

The fiber fuse effect is a complex process that can be roughly divided into several stages. The first is the optical breakdown initiation in the fiber core, which can be caused by various external factors, for example, fiber heating, electrical discharge, or contamination. The second is the emergence and propagation of a self-sustaining plasma focus (spark) along the fiber core. For a stable observation of the fuse effect, the laser power density at both stages of the process must be sufficient to create and maintain the necessary temperature in the plasma spark, and the size of the plasma spark must be comparable to the diameter of the fiber core.

The optical breakdown effect is caused by the nonlinear absorption of optical radiation in the fiber core. It directly depends on the type of fiber and its degree of doping. For example, the breakdown of fibers with pure quartz core requires a power density exceeding 40 MW/cm$^2$, while fibers with a germanium admixture require a threshold breakdown power density of 1 MW/cm$^2$. The centers of nonlinear radiation absorption or "coloring centers" are point defects in the fiber core or E′ centers, which are associated with oxygen holes in the X-Si bonds, where Ge germanium or Si silicon can replace X. It is noted that the concentration of GeE′ centers is preserved when the temperature increases, while the concentration of SiE′ begins to decrease at 573 K to complete disappearance at 973 K [9].

Chemically, the optical breakdown process can be described as follows: in the first step, the compound Ge-O-Si decomposes into molecular oxygen $O_2$ and the compound Ge-Si under the action of high temperature. Then, the new compound is ionized, forming a positively charged Ge-Si$^+$ ion and an electron e$^-$. Thus, during thermal ionization, the electrons released by the reaction begin to move in the fiber core and form an avalanche,

the bonds between the atoms are torn, and plasma is formed. In this case, the concentration of germanium-silicon ions and the released electrons are the same.

This work is devoted to the construction of a thermophysical mathematical model of an optical breakdown initiation stage at the fiber end in the presence of an initiator in the form of a metal plate. The optical breakdown of the fiber end SMF-28e pressed to duralumin plate D16 according to GOST RF 4784-97 was studied. The process of thermal source development in the fiber core was investigated, and the conditions leading to the appearance of plasma spark were determined. Based on numerical calculations and available experimental data, the authors have suggested that plasma spark motion starts at the moment when the temperature front of plasma formation reaches the fiber core–shell boundary. In this case, the center of plasma formation has a temperature of more than 5000 K. Estimates were made of the plasma spark growth time to the size of the fiber core. For example, at the radiation power of 0.5 W and wavelengths $\lambda_0 = 1080$ nm and $\lambda_0 = 1550$ nm, this time was 1.7 μs and 5 μs, respectively. When the power of the laser radiation causing the optical breakdown increased to 3 W or more, the difference between the times for different wavelengths decreased. It can be seen from the results of the calculations that at radiation powers of 10 W, the times for reaching the fiber core boundary by the plasma focus lie in a narrow interval from 0.09 μs for $\lambda_0 = 1080$ nm to 0.18 μs for $\lambda_0 = 2050$ nm.

Thanks to the mathematical simulation of the fuse effect, it is possible to better study the mechanisms of its emergence, evolution, and transients between the stages of the process. Using the Comsol software package, it was possible to implement an algorithm for solving the model. We used the constant Newton method of solving linear equations, with a Jacobian update at each time step, to solve the emerging nonlinear equations. The solver parameters were chosen to ensure minimal solution time and acceptable convergence. All parameters were described in Section 2 (Mathematical Model of Plasma Spark Initiation). The system of equations was solved in a multi-threaded mode using PARDISO Solver. The connected Andersen acceleration allowed for a faster and improved solution convergence. The main improvement of this model is the preliminary introduction of boundary conditions and the search for the two-dimensional shape of the heat source. This allowed us to obtain physically correct temperature data, which are confirmed by experimental and literature data. With the selected parameters, the solution time of the model was 22 s. The results of this work will form the basis of the model of plasma spark propagation along the fiber and will be combined with it. The process of plasma spark propagation along the fiber has not been studied within the framework of this work.

When using lasers with different wavelengths, different times are needed to initiate the optical breakdown process. This time is shorter the closer the laser wavelength is shifted to the ultraviolet part of the spectrum. This is not surprising because decreasing the wavelength (increasing the frequency) leads to an increase in the energy carried by the radiation quantum. Therefore, laser systems for glass micromachining operating in the ultraviolet or visible range have found wide applications. Due to the high probability of optical breakdown in quartz, there is a limitation associated with the use of full-body quartz optical fibers for the transmission short-wave part of the electromagnetic radiation range beginning with the ultraviolet. Hollow fibers are used to transmit radiation from ultraviolet to $\gamma$.

In this work, the minimum radiation power required to initiate optical breakdown was estimated. It was shown that this power increases with increasing wavelength. This circumstance suggests that, first, the quartz fiber core has the lowest absorption in the infrared region of the spectrum, and second, the laser radiation more easily ionizes the fiber material when shifting to the ultraviolet region. Thus, the presence of the minimum power required to trigger the fiber fuse effect tells us that there is a red border at the beginning of the process. This fact is confirmed by the publications of other authors. For example, the article by S. Todoroki [8] reported that the minimum start time of the breakdown process observed in the experiment was 0.18 s, and the minimum power at which it was possible to

start the reaction in the SMF-28 fiber was 0.4 W. In the future, it is planned to investigate the red border of the reaction occurrence and experimentally confirm the simulation results.

The maximum temperature $T_{max}$, which occurs at the center of the optical spark, also has a dependence on laser radiation wavelength. Calculations have shown that $T_{max}$ increases linearly with increasing optical radiation power and decreases with increasing wavelength. In addition, the growth rate of maximum temperature also depends on the wavelength. According to calculations, temperatures of several tens of thousands or even hundreds of thousands of kelvins occur at the plasma droplet center. For example, for $\lambda_0$ = 1080 nm at a laser radiation power of 10 W, the center of the plasma droplet heats up to temperatures over 500,000 K. However, such high-temperature values are localized in the center of the plasma focus and decrease to 5000 K closer to the fiber core–shell interface. In turn, the outer layers of the fiber are practically not heated via the plasma spark. That is, the optical breakdown area is very small. The values obtained qualitatively agree with the results of [6], where the maximum temperature inside the fiber reached 450,000 K for radiation with wavelength $\lambda_0$ = 1550 nm and power $P$ = 2 W.

The presented results of numerical calculations have shown that the minimum temperature in the center of the plasma region, at which the condition for the transition of optical breakdown to the plasma spark motion process is satisfied, is approximately 14,500 K. The value of this temperature is almost the same for different wavelengths. At temperatures below the minimum temperature, the temperature front of plasma formation does not reach the fiber core–shell boundary. In such a case, optical breakdown occurs, but it is not transformed into the process of plasma spark propagation. The plasma spark does not gain enough critical mass to begin the process of self-sustaining propagation. In experiments, this often leads to the melting of the fiber end and its glow or to the destruction of the waveguide section.

## 6. Conclusions

A mathematical model describing the stage of optical breakdown in a single-mode optical fiber was developed in this work. It was built in Comsol Multiphysics, Heat Transfer in Solids (ht) Interface. The problem was solved in multi-threaded mode using the PARDISO Solver. The computation of the algorithm took 22 s.

The boundaries of temperature fronts corresponding to the regions where different phase states of quartz glass can occur were determined, namely liquid, gaseous, and plasma. The time required for plasma spark formation at different powers and wavelengths of optical radiation was estimated by reaching the temperature front of plasma formation at the core–shell boundary.

As a result of calculations for radiation of different wavelengths, the minimum values of the radiation powers required for the initiation of the fuse effect were obtained. For these powers, the maximum temperatures in the plasma spark were found. It is shown that the peak temperature values are localized near the center of the breakdown initiation region. Toward the edges of this region, the temperature decreases rapidly. Calculations showed a linear growth of the maximum temperature in the breakdown region with increasing radiation power. At the minimum powers of optical radiation, the plasma spark initiation time for different wavelengths is approximately the same and is equal to 1 ms. With increasing radiation power, the initiation time becomes different for different wavelengths and decreases with increasing power according to the law close to the exponential.

The results can be used to predict the minimum power of the optical radiation injected into the fiber required for the occurrence of optical breakdown in the fiber at different wavelengths. The mathematical model formulated in this work allows for estimating the start time of the fiber fuse effect (plasma spark initiation) as well as the radial and longitudinal dimensions of the plasma spark and other fiber heating regions. To improve the predictive properties of the model and refine the calculation results, a more correct accounting of the thermal–physical properties of the fiber material in the regions where

different phase states of the substance are realized. In addition, it makes sense to consider the case of non-ideal contact of the fiber with the metal plate.

**Author Contributions:** V.A.S. and A.Y.P. wrote the main manuscript text; V.A.S., A.V.P. and Y.A.K. developed the model; Y.A.K. prepared Figures 1, 2, 4, 6 and 8; V.A.S. prepared Figures 3, 5, 7 and 9–11; A.V.P., S.S.A. and A.A.P. supervised the research and verified the model data. All authors have read and agreed to the published version of the manuscript.

**Funding:** This study was supported by a grant from the Russian Science Foundation No. 23-21-00169: https://rscf.ru/en/project/23-21-00169/ (accessed on 1 January 2023).

**Data Availability Statement:** Not applicable.

**Conflicts of Interest:** The authors declare no conflict of interest.

## Appendix A

**Table A1.** Thermophysical parameters of the materials used in the model.

| Thermophysical Parameters: Density, Specific Heat, Thermal Conductivity | | | |
|---|---|---|---|
| **Core and Shell—Fused Quartz** | | | |
| **Parameter** | **Temperature Range, *K*** | | **Equation** |
| $\rho,\frac{kg}{m^2}$ | | <80 | 2220.007 |
| | 80 | 1000 | $2219.390 + 0.011T - 4.708\cdot10^{-5}T^2 + 6.981\cdot10^{-8} - 4.995\cdot10^{-11}T^4 + 1.431\cdot10^{-14}T^5$ |
| | | >1000 | 2217.536 |
| $C_p,\frac{J}{kg\cdot K}$ | | <10 | 4.350 |
| | 10 | 130 | $-8.361 + 0.814T + 0.050T^2 - 4.462\cdot10^{-4}T^3 - 1.681\cdot10^{-6}T^4 - 2.285\cdot10^{-9}T^5$ |
| | 130 | 925 | $62.309 + 1.926T + 0.004T^2 - 1.704\cdot10^{-5}T^3 + 1.881\cdot10^{-8}T^4 - 7.062\cdot10^{-12}T^5$ |
| | 925 | 2000 | $890.996 + 0.370T - 1.107\cdot10^{-4}T^2 + 3.139\cdot10^{-8}T^3$ |
| | | >2000 | 1440.600 |
| $k,\frac{W}{m\cdot K}$ | | <5 | 0.109 |
| | 5 | 280 | $0.092 + 0.003T + 4.343\cdot10^{-5}T^2 - 2.258\cdot10^{-7}T^3 + 3.049\cdot10^{-10}T^4$ |
| | 280 | 1400 | $-0.985 + 0.018T - 5.290\cdot10^{-5}T^2 + 7.552\cdot10^{-8}T^3 - 5.008\cdot10^{-11}T^4 + 1.311\cdot10^{-14}T^5$ |
| | 1400 | 1473 | $-1798.509 + 4.192T - 0.003T^2 + 8.408\cdot10^{-7}T^3$ |
| | | >1473 | 10.450 |
| **Polymer protective coating—acrylate** | | | |
| $\rho,\frac{kg}{m^2}$ | | | 1190 |
| $C_p,\frac{J}{kg\cdot K}$ | | | 1420 |
| $k,\frac{W}{m\cdot K}$ | | | 0.19 |
| **Metal plate activator—Duralumin plate D16 of the GOST 4784-97 according to 2024 of the ISO** | | | |
| $\rho,\frac{kg}{m^2}$ | | <0 | 2813.900 |
| | 0 | 755 | $2813.898 + 0.028T - 7.443\cdot10^{-4}T^2 + 1.039\cdot10^{-6}T^3 - 5.689\cdot10^{-10}T^4$ |
| | | >755 | 2673.500 |

**Table A1.** *Cont.*

| Thermophysical Parameters: Density, Specific Heat, Thermal Conductivity | | |
|---|---|---|
| **Core and Shell—Fused Quartz** | | |
| **Parameter** | **Temperature Range, K** | **Equation** |
| $C_p$, $\frac{J}{kg \cdot K}$ | <116 | 564.860 |
| | 116     700 | $198.819 + 3.941T - 0.007T^2 + 5.218 \cdot 10^{-6}T^3$ |
| | >700 | 1129.750 |
| $k$, $\frac{W}{m \cdot K}$ | <4 | 3.035 |
| | 4     50 | $-0.676 + 0.938T - 0.003T^2$ |
| | 50     120 | $10.259 + 0.616T - 8.031 \cdot 10^{-4}T^2$ |
| | 120     700 | $-12.173 + 1.076T - 0.003T^2 + 7.721 \cdot 10^{-6}T^3 - 5.396 \cdot 10^{-9}T^4$ |
| | >700 | 171.554 |

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
