# Peer review of "Mathematical Model of Fuse Effect Initiation in Fiber Core"

_algorithms, doi:10.3390/a16070331_

Round 1

Reviewer 1 Report

The paper entitled "Mathematical model of fuse effect initiation in fiber core" discloses a mathematical model numerically solved to analyze fuse effect in optical fiber. It is well written and interesting but need some revision. Here are my comments.

Comments

1. Use SI unit conventions: s and not sec for second for example. Mobility is in cm^2/(Vs) and not as stated in line 155, kelvin and not degree Kevin, ...

2. Give the values of all the parameters used in the simulations (thermal conductivity, specific heat, ... for the different materials).

3. In table 1, the fiber core and the fiber shell are identical material, but the core of SMF-28 is Ge-doped.

4. Explain what you mean by standard Gaussian distribution \Gamma in the text and relation (3).

5. You mention in the text that energy is nonlinearly absorbed (line 128) but the fiber absorption coefficient \alpha also contains a linear term. Better explain.

6. There are some strange signs (cyrillic characters?) in lines 12& and 172.

7. Be consistent with the number of digits for the numbers given in the table and in the text.

8. In relation (&0), \Gamma is decomposed in \Gamma(r)*\Gamma(z) but without any explanation.

9. Line 220: what is physical millisecond?

10. Table 3: write 'Concentration' and 'Concentarion'.

11. Use a dot for the decimal separator and not a comma (figures !, 10

12. Line 305: "At high powers there is a significant difference in the initiation times (see Fig. 8)." It seems to be the opposite in figure 8.

13. Line 392: " All parameters were specified in Chapter 3." What is chapter 3. Moreover, the material parameters are not given (see question 2).

14. Line 398: "But the possibilities of realization of this model and the software allow to improve the accuracy of the solution or to apply other methods to estimate the data." Please elaborate this part as the uncertainty is not tackle for the presented model.

15. Line 364: O2 and not O2.

16. Reference 17 has no year field.

Author Response

Hello, dear reviewers. Thank you very much for your detailed and truthful review.  My colleagues and I have taken all your comments into account and made corrections to the manuscript. All changes are highlighted in green in the text.

1) All units in the text have been converted to standard SI form. Lines 60, 172, 177,267, 357,448,457

2)Since thermophysical parameters have a complex dependence on temperature, and are difficult to display in the article, it was decided to refer to the reference data presented in the Comsol Multyphisycs library. Line 132-134

3) The mistake has been corrected. Line 120

4) An explanation of the standard Gaussian distribution has been given. Lines 151-152

5) Removed the mention " nonlinear" in line 147 (previously 128). The absorption index is complex and has linear and non-linear components, the explanation is given in lines 156-162

6) The characters have been changed to the Latin alphabet. Line 139,194

7) Tabular and text variables were checked and matched.

8) An explanation of the decomposition of the Gaussian distribution has been added, it becomes two-dimensional. Line 154-155

9) The word physical was removed, it denoted the time of the real process being modeled. Line 244

10)The word has been corrected. Line 267-268

11) The figures have been replaced. Lines 317-318, 343-344

12) Replaced the sentence with the correct one. Line 330-331

13) We put a link to the section of the article "2. Mathematical model of plasma spark initiation. Line 420

14) The sentence was removed. The proposal was not removed from the final version of the article due to an oversight by the authors. The sentence supposed flexibility in adjusting the functionality of the model, but it also gave uncertainty in the conclusions. Line 426

15)Replaced the index with the correct one. Line 392

16)We replaced the reference with the correct one. Lines 557-559

Reviewer 2 Report

The authors proposed a Mathematical model of fuse effect initiation in fiber core.  The theme is interesting but the following corrections are needed for the final publication.

1.                  In the abstract, the real life application of this work is missing. I will encourage the authors to add this.

2.                  In the introduction, the background study presentation is very poor. Authors are encouraged to improve the technical writing.  

3.                  Some acronym is not clearly discussed before use.

4.                  In methods section, the study is not clear for the new readers. Need to improve it clearly. Authors should care about it.

5.                  There is no proper explanation of equation in this manuscript. Need to add those for the better understanding of the new readers.

6.                  Provide a more numerical explanation of the investigated parameters.

7.                  There are many grammatical mistakes. Authors should take care of it.  

8.                  The comparisons table is totally avoided by the authors.

9.                  Figures quality is very poor. Authors should care about it.

There are many grammatical mistakes. Authors should take care of it.  

Author Response

Hello, dear reviewers. Thank you very much for your detailed and truthful review.  My colleagues and I have taken all your comments into account and made corrections to the manuscript. All changes are highlighted in green in the text.

1) In the abstract and introduction, where appropriate, we have added information on possible practical applications of quasi-periodic structures resulting from fiber breakdown. Lines 10-15

2)The authors have tried to present as fully as possible the history of the development of research on optical fiber breakdown and have tried to refer to the most significant works in this field. If the esteemed reviewer is aware of specific works missed by the authors, they would be very grateful if the reviewer could point them out.

3) We have tried to explain all the abbreviations used.

4) An example of using the COMSOL package to solve finite element problems and the packages used for this publication have been described. Lines 206-217
A description of the methods used to solve the problem has been added. Lines 282-294

5) We have tried to explain in more detail the equations used, interfacial boundary conditions and boundary conditions. If this does not seem enough to the esteemed reviewer, we would be glad to get an explanation of what else should be written to explain the equations used. Lines 135-136

6) Since thermophysical parameters have a complex dependence on temperature, and are difficult to display in the article, it was decided to refer to the reference data presented in the Comsol Multyphisycs library. Line 132-134

7) The authors have more carefully read the text of the paper and tried to correct all grammatical errors.

8) The authors tried as much as possible to compare the results obtained at different wavelengths and input signal powers, which is shown in Figures 8 and 10, as well as in Table 4. It is not quite clear what the reviewer meant by the term "the comparisons table". If the respected reviewer would explain what he meant in note 8, the authors would be grateful and would take the note into account.

9) The authors have tried to refine the drawings and improve the quality of the figures.

Round 2

Reviewer 1 Report

Most of my comments have been processed.

Major issues:

1. In relation 4 why is it a sum and not a multiplication?

2. "Line 132: "Since these functions have a complex form that depends on the temperature, it is not appropriate to include them in this article. As mentioned above, all thermophysical properties of materials are taken from the COMSOL Multiphysics library." ==> I cannot agree with this sentence. A paper should present material that is reproducible by others (by COMSOL or other tools) . So give enough detail on the material parameters used in this study. It is not sufficient to refer to COMSOL library.

Minor issues:

1. Line 158: M^-1 instead of m^-1

2. line 177: 6,3 instead of 6.3

3. Table 2: 3 or 4 decimal places for n. Use 4 places for n everywhere in the table.

Author Response

Major issues:

Hello, esteemed reviewer. Thank you for reviewing our publication. We have tried to take your comments into account as much as possible and make corrections to the article.

  1. In relation 4 why is it a sum and not a multiplication? A: Corrected the sign to multiplication, you are right, multiplication is more correct here
  2.  "Line 132: "Since these functions have a complex form that depends on the temperature, it is not appropriate to include them in this article. As mentioned above, all thermophysical properties of materials are taken from the COMSOL Multiphysics library." ==> I cannot agree with this sentence. A paper should present material that is reproducible by others (by COMSOL or other tools) . So give enough detail on the material parameters used in this study. It is not sufficient to refer to COMSOL library.
    A: Corrected the reference to line 132 line. Added table of thermophysical parameters to Appendix A, Table A1.

    Minor issues:

    1. Line 158: M^-1 instead of m^-1
      A: Corrected to m^-1
    2. line 177: 6,3 instead of 6.3
      A: Corrected 
    3. Table 2: 3 or 4 decimal places for n. Use 4 places for n everywhere in the table.

    A: Corrected refractive index values

Reviewer 2 Report

The manuscript might be accepted in its current form right away.

Moderate English correction needed

Author Response

Hello, esteemed reviewer. Thank you for reviewing our publication. We have tried to take your comments into account as much as possible and make corrections to the article.

The text of the article has been carefully proofread and the authors have tried to improve the English of the article as much as possible.

Round 3

Reviewer 1 Report

Corrections have been done.

Paper is now suitable for publicaion.